# Empathic Accuracy in Chronic Pain: Exploring Patient and Informal Caregiver Differences and Their Personality Correlates

**DOI:** 10.3390/medicina55090539

**Published:** 2019-08-27

**Authors:** Carlos Suso-Ribera, Verónica Martínez-Borba, Alejandro Viciano, Francisco Javier Cano-García, Azucena García-Palacios

**Affiliations:** 1Department of Personality, Assessment, and Psychological Treatments, Jaume I University, 12071 Castellón de la Plana, Spain; 2Department of Personality, Assessment, and Psychological Treatments, University of Seville, 41018 Seville, Spain

**Keywords:** empathic accuracy, chronic pain, informal caregiver, spouse, health status, personality

## Abstract

*Background and objectives:* Social factors have demonstrated to affect pain intensity and quality of life of pain patients, such as social support or the attitudes and responses of the main informal caregiver. Similarly, pain has negative consequences on the patient’s social environment. However, it is still rare to include social factors in pain research and treatment. This study compares patient and caregivers’ accuracy, as well as explores personality and health correlates of empathic accuracy in patients and caregivers. *Materials and Methods:* The study comprised 292 chronic pain patients from the Pain Clinic of the Vall d’Hebron Hospital in Spain (main age = 59.4 years; 66.8% females) and their main informal caregivers (main age = 53.5 years; 51.0% females; 68.5% couples). *Results:* Patients were relatively inaccurate at estimating the interference of pain on their counterparts (*t* = 2.16; *p* = 0.032), while informal caregivers estimated well the patient’s status (all differences *p* > 0.05). Empathic accuracy on patient and caregiver status did not differ across types of relationship (i.e., couple or other; all differences *p* > 0.05). Sex differences in estimation only occurred for disagreement in pain severity, with female caregivers showing higher overestimation (*t* = 2.18; *p* = 0.030). Patients’ health status and caregivers’ personality were significant correlates of empathic accuracy. Overall, estimation was poorer when patients presented higher physical functioning. Similarly, caregiver had more difficulties in estimating the patient’s pain interference as patient general and mental health increased (*r* = 0.16, *p* = 0.008, and *r* = 0.15, *p* = 0.009, respectively). Caregiver openness was linked to a more accurate estimation of a patient’s status (*r* = 0.20, *p* < 0.001), while caregiver agreeableness was related to a patient’s greater accuracy of their caregivers’ pain interference (*r* = 0.15, *p* = 0.009). *Conclusions:* Patients poorly estimate the impact of their illness compared to caregivers, regardless of their relationship. Some personality characteristics in the caregiver and health outcomes in the patient are associated with empathic inaccuracy, which should guide clinicians when selecting who requires more active training on empathy in pain settings.

## 1. Introduction

According to epidemiological studies, chronic pain, which is defined as recurrent pain lasting for long periods of time (i.e., over three months according to the literature) [1], is becoming alarmingly frequent. Specifically, this disease is currently estimated to affect between 20 and 30% of the population globally [2,3,4,5] and, with the age distribution changing towards the elderly, the incidence of chronic pain is likely to raise in the coming years due to the increased prevalence of this disease in the elderly [6].

Historically, chronic pain research and theories have embraced different perspectives, from clearly reductionist considerations to complex multidimensional approaches to the pain experience [7]. The more traditional view considered pain as a symptom of illness and assumed that pain severity was proportional to the amount of tissue damage. This reductionist approach has now been substituted by a more comprehensive view of pain as a complex and unpleasant sensory and emotional experience [8] that can only be understood as an interplay between sensory, cognitive, emotional, and social components [9]. As a consequence of this, a biopsychosocial approach to pain is currently considered the gold standard model to understand the pain experience [10,11,12] and interdisciplinary treatments of chronic pain have become the recommended approach to this disease [10,13,14].

Indeed, important milestones have been reached in the development and incorporation of the psychological perspective in this biopsychosocial approach to pain and the importance of psychological therapies for pain management is well-supported [15,16,17]. The literature investigating the role of the “social” part has also received attention, but not to the same extent. There is evidence to suggest that environmental elements, such as social support [18,19,20,21] and spouse-patient interactions [22,23,24,25], are important factors associated with patient outcomes, including functioning despite pain, emotional well-being, and quality of life, among others. 

Another interesting line of research into the social determinants of patient adjustment to pain is the study of empathic accuracy, which has some tradition in the chronic pain literature. Empathic accuracy is conceptualized as an individual’s interpersonal ability to judge the other person’s status and differs from the traditional affective form of empathy [26]. Research into empathic accuracy can be included in a tradition of studies exploring the experience of pain from a caregiver’s perspective, which includes not only spouse appraisals about the other’s status (i.e., empathic accuracy) but also perceptions about one’s status (i.e., caregiving burden) and emotional and behavioral reactions to pain (e.g., validating or critical responses) [27,28]. Two types of empathic accuracy have been traditionally tested, both relying on self-reports. On the one hand, correlational accuracy indicates how well self-reported patient pain is associated with the observer’s perception of a patient’s pain. This perspective is frequently used when patient and observer ratings are based on different response scales, but its interest is limited because it does not provide information about the size and the direction of inaccuracy. An alternative to this is the paired comparison accuracy, which evaluates the extent to which observers are accurate at estimating the other’s status (i.e., overestimation and underestimation) [29]. As noted earlier, this requires the use of the same response scale in patients and observers, but provides more detailed information about empathic accuracy. Thus, the latter will be preferred in the present investigation.

In the chronic pain field, empathic accuracy has been usually defined as the caregiver’s or, more frequently, the spouse’s accuracy to decode the patient’s pain severity; it. is considered to be an important interpersonal process in the patient’s adaptation to chronic pain [28]. Accordingly, most research into empathic accuracy in chronic pain settings has focused on the exploration of health and adaptation correlates of empathic accuracy in a couple’s context. So far, the literature suggests that spouses are quite accurate in estimating their partner’s pain levels [30] irrespective of spousal sex characteristics [31] and supports the aforementioned idea that a spouse’s accuracy in estimating a patient’s pain levels is associated with positive outcomes in the patient, such as emotional well-being [32,33,34] and patient satisfaction with spousal support [35]. A recent meta-analysis also indicated that informal caregivers are more accurate than physicians and revealed that pain is more frequently underestimated in male patients [29].

The existing literature that has looked into the relationship between empathic accuracy and patient outcomes is encouraging, but important questions in this field remain unanswered. For instance, spouse and patient predictors of accuracy in the field of pain have rarely been investigated. So far, some examples of spousal characteristics that have been related to (poorer) empathic accuracy are depression in the spouse, being a male spouse rating a female patient, and reporting high levels of spousal catastrophizing [36,37]. An additional gap in pain literature is that the most research has focused on spouses as important sources of interaction, while the role of other relatives and even friends that are known to play a central role in informal caregiving in chronic patients [38] have been ignored. In chronic pain, because the prevalence of the disease raises dramatically with age [6], being a widow or being in a relationship with a partner that is also disabled is frequent [39], which boosts the need for informal caregivers other than spouses and makes the study of empathic accuracy in populations other than spouses important. Finally, empathic accuracy has traditionally been conceptualized from a patient’s perspective (i.e., patient as a recipient of empathy). Because pain is also known to impact negatively on the family [40], investigating the patient’s ability to estimate the consequence of pain on their significant others might also render important service to pain literature. 

The present study aims to contribute to the social aspect of biopsychosocial literature by exploring how chronic pain correlates with empathic accuracy for both patienst and informal caregivers, including spouses and other main caregivers. According to Goubert et al. [28], empathic accuracy is determined by contextual factors (e.g., type of relationship and affinity), top-down mechanisms (e.g., the observer’s personality characteristics and past learning experiences), and bottom-up influences (e.g., the observed person’s characteristics). In the present study, contextual, top-down and bottom-up mechanisms will be included. Specifically, the explored correlates of empathic accuracy will include the personality characteristics of patients (i.e., bottom-up factor) and informal caregivers (i.e., top-down factor) because there is a large tradition of research showing a relationship between personality and empathy in social psychology research, but not in chronic pain [31,37,41]. While many personality characteristics exist, we chose to evaluate the five core dimensions of the Five Factor Model of personality because this model is cross-culturally robust and has been repeatedly and reliably associated with important study variables (i.e., health status and empathy) across different populations [42,43], including chronic pain [44]. In doing so, we will also explore the association between empathic accuracy and patient (i.e., bottom-up factor)/caregiver (i.e., top-down factor) health status in order to investigate whether existing findings are replicated in our sample. Specifically, we expect to replicate, for empathic accuracy in the patient’s pain severity and interference, the correlations between certain personality traits (i.e., agreeableness and openness to experience) and empathy and to find empathic accuracy in the patient to be associated with the patient’s health status. We also anticipate that informal caregivers will be quite accurate in estimating a patient’s pain and interference levels. Patients’ empathic accuracy in relation to the pain’s impact on the caregiver, patient and caregiver empathic accuracy differences as a function of the type of relationship (i.e., contextual factor), and the relationship between patient and spouse empathic accuracy and the caregivers’ health status will be investigated in an exploratory manner. 

Ultimately, with the current investigation, we expect to help guide interdisciplinary interventions in a more effective manner by exploring social factors that could be incorporated in current treatments for pain. Including empathic accuracy training in such interventions would be beneficial for a number of reasons. Empathic accuracy is argued to lead to prosocial and support-related behaviors [45], proximity feelings in a relationship [46], and adaptive solution of conflicts in romantic relationships [47]. In fact, empathic accuracy has been argued to have a survival value for the patient as it would enhance assistance from others in the presence of pain (i.e., affective and behavioral responses) [28,31]. On the contrary, poor estimation by spouses might lead to inadequate support from a caregiver [48], which might ultimately exert a negative impact on a patient’s ability to manage pain [32]. By exploring contextual, bottom-up, and bottom-down correlations of empathic accuracy, the present study aims at contributing to the important existing literature on social factors involved in the chronic pain experience.

## 2. Materials and Methods

### 2.1. Sample and Procedures

In this study, potential participants were patients with a prospective appointment from the beginning of 2014 to the end of 2015 at the pain clinic of a tertiary hospital in Spain (Vall d’Hebron Hospital in Barcelona) and their informal caregivers. Participants were recruited via postal mailing. A letter was sent to patients’ home approximately one month prior to their medical consultation at the pain clinic. The letter included two protocols: one for the patient and one for the main informal caregiver. Each protocol included a study information sheet, an informed consent form, and questionnaires. Patients and informal caregivers were asked to complete the questionnaires separately and were given two additional envelopes. The completed questionnaires had to be returned separately in the corresponding sealed envelope the day of the prospective consultation, together with the signed informed consent forms, either to a physician or to the lead researcher, Carlos Suso-Ribera. Participants were officially enrolled at this stage by collecting the informed consent form and ensuring that eligibility was met. 

In total, 335 dyads were willing to participate and returned the signed informed consent form. However, when revising the questionnaires, 81 protocols were found to contain incomplete data. Missing information from 38 of these dyads could be obtained with phone calls, but 53 cases were lost due to difficulties in contacting informal caregivers. As a result of this, the final sample with complete data was composed of 292 patients and their main informal caregivers. 

Inclusion criteria for patients included experiencing recurrent pain over the last three months (i.e., chronic pain), being aged 18 or over, completing the questionnaires separately from the caregiver, and providing written informed consent to participate. Eligibility criteria for informal caregivers were being over 18 years of age, acting as the main informal caregiver for the patient, completing the questionnaires separately from the patient, and returning the written informed consent. We verified that caregivers were indeed the main informal caregivers when the patient or both returned the questionnaires during the medical appointment. Because the information letter was very clear at this stage, all patients confirmed that their main caregiver responded to the questionnaires. We also verbally confirmed that the questionnaires were completed separately and ensured that the questionnaires were returned in separate and sealed envelopes. 

The Ethics Review Committee of the Vall d’Hebron Hospital in Barcelona approved the present study and all its procedures (PR(ATR)59/2010; approval date 8 May 2010).

### 2.2. Measures

Pain severity and pain interference in the last 24 h were measured with a numerical rating scale (NRS) ranging from 0 = no pain/interference to 10 = worst possible pain/interference, which was adapted from the Brief Pain Inventory [49]. Regarding pain severity, patients were asked to rate their own pain and informal caregivers were requested to estimate the pain of the patients. Pain interference was rated by both the patient and the caregiver as perceived interference on oneself and estimated interference of the patient’s pain on the other.

The personality of patients and informal caregivers was evaluated with the NEO-Five-Factor-Inventory (NEO-FFI) [50,51], which evaluates the five core traits of the Five Factor Model of personality, namely neuroticism, extraversion, openness to experience, agreeableness, and conscientiousness. This comprehensive personality model is the most widely used in different settings [42], including chronic pain [44]. Neuroticism refers to a tendency to experience negative emotional states, such as depression, anxiety, and anger. Extraversion reflects a predisposition to be talkative, lively, enthusiast, and dynamic. Openness to experience characterizes curious, untraditional persons with a variety of interests and high sensitivity to arts. Agreeableness defines altruistic, honest, and helpful individuals. Finally, high conscientiousness scores are obtained by individuals prone to hard work, achievement-directed, and meticulous [50]. The Five Factor Model comprises 60 statements (12 per personality dimension) using a 5-point Likert scale that range from 0 = totally disagree to 4 = completely agree. For each personality dimension, scores range from 0 to 48. The NEO-FFI has obtained good internal consistency indices in previous (0.66 < *α* < 0.81) [51], as well as in the present study for both patients (0.70 < *α* < 0.85) and informal caregivers (0.67 < *α* < 0.84).

The health status of patients and informal caregivers was assessed with the Short Form-36, which contains 36 items referring to eight dimensions of an individual’s perceived health [52,53]. These include physical functioning (i.e., the ability to perform in daily activities), physical role (i.e., the capacity to perform in work-related duties), bodily pain (i.e., average pain severity in the last four weeks), general health (i.e., perception of present and future health), vitality (i.e., perceived personal energy), social functioning (i.e., interference of health in interpersonal relationships), role emotional (i.e., interference of emotions on functioning), and mental health (i.e., overall mental well-being) [54]. Two composite scores (i.e., overall physical and mental health) can be computed with the aforementioned subscales [53]. However, the use of subscales was preferred in the present study due to a number of reasons. First, because being retired or unemployed is frequent in chronic pain populations [55], including the present one, the physical role scale is of little relevance. Additionally, the use of the physical composite score would also be problematic because it would contain the measure of bodily pain, which would contaminate the relationship between the dependent variable (i.e., empathic accuracy in pain severity) and the independent variable (i.e., physical health status). To minimize the number of statistical comparisons, which is important to reduce type I errors, we selected three of the eight indicators of patient and spouse health status that represent different dimensions of perceived health, namely physical functioning, general health, and mental health [54]. In the Short Form-36, all scales are standardized and scores range from 0 to 100, with higher rating indicating better health. Previous research had evidence good internal consistency estimates of these scales (0.78 < α < 0.94) [52], which was replicated for both patients (0.78 < α < 0.90) and informal caregivers (0.87 < α < 0.95) in the present investigation.

### 2.3. Data Analysis

The analytic strategy included a descriptive analysis of the sample, followed by an analysis of differences in patient and caregiver empathic accuracy by means of Student *t*-tests and a series of Pearson bivariate correlations between patient and caregiver empathic accuracy, personality, and health status both from the patient and the caregiver perspective. Specifically, differences in patient and caregiver empathic accuracy were investigated between patients and caregivers and between partners and non-partners, as well as between men and women because the existing literature shows the importance of sex in the pain experience [56,57].

Several measures of patient and caregiver empathic accuracy were calculated, which is in line with past research [58]. First, we calculated the disagreement in patient pain severity (patient self-perceived pain severity and informal caregiver’s perception of patient pain severity), disagreement in patient pain interference (patient self-perceived interference of pain and informal caregiver’s perception of pain interference on the patient), and the disagreement in informal caregiver pain interference (informal caregiver self-perceived interference of pain and patient’s perception of pain interference on the caregiver). In all cases, a positive score reflected underestimation of the other’s status, while a negative value were interpreted as overestimation of the other person’s status. These three variables will be the main outcomes in the study. However, we calculated three additional variables for each of these three outcomes (i.e., disagreement in patient pain severity, disagreement in patient pain interference, and disagreement in informal caregiver pain interference). Namely, the three additional variables were the absolute value of each difference (i.e., poor estimation irrespective of the direction of the disagreement) and a dichotomous variable for overestimation and underestimation (where “0 = disagreement lower than 1 *SD* from the average score” and “1 = estimation with a disagreement of over 1 *SD* from the average disagreement score above and below the mean, respectively”). These additional outcomes were used in the correlation analyses to facilitate the interpretation of the relationship between patient and caregiver empathic accuracy and personality and health status. Finally, a linear multivariate regression analysis was computed to explore the unique associations between study predictors and the three main outcomes (i.e., empathic accuracy on patient pain severity and interference and empathic accuracy on caregiver pain interference). For each of the three regressions, the multivariate regression included sex and the patient and spouse personality and health characteristics that are found to significantly correlate with study outcomes.

Because of the large number of comparisons made in the Pearson correlation analyses, a more conservative *p* value of 0.01 was used as a cut-off for the interpretation of the bivariate associations to minimize the risk of type I errors and to reduce overestimation of unimportant correlations. The new alpha level was calculated using a Holm-Bonferroni sequential correction, a less restrictive correction than the original Bonferroni correction [59]. Because multiple comparison corrections often leads to rare occurrences of significant values, effect sizes (Cohen’s *d*) were reported to facilitate the interpretation of mean differences analyses (*t*-tests) and the strength of the bivariate correlations (*r* values) and multivariate regressions (standardized *β* coefficients and explained variance) will be discussed too [60]. All analyses were conducted with SPSS version 25 [61].

## 3. Results

### 3.1. Sample Characteristics

Table 1 shows the demographic characteristics of the sample. In total, 584 individuals provided complete dyadic data (292 patients and 292 main informal caregivers). The majority of patients were characterized by experiencing musculoskeletal pain, mostly in the low back and the neck, but patients with any type of non-cancer pain were included in the study (i.e., to obtain a representative sample of the population treated at a specialized pain clinic). An analysis of sex distribution revealed that the majority of patients were females (66.8%), while men and women were similarly represented in the sample of informal caregivers (51.0% of females and 49.0% of males). The most frequent relationship between patients and informal caregivers was being a couple (68.5%). Other frequent relationships included being the patient’s child (20.2%), being a sibling (4.1%), and being the patient’s parent (2.7%). Overall, patients were slightly older than informal caregivers (59.35 years vs. 53.52 years, *t* = 4.63, *p* < 0.001). Informal caregivers had a higher educational level (25.3% of patients and 34.6% of informal caregivers had more than 12 years of education, *χ*^2^ = 16.60, *p* < 0.001) and were more likely to be working at the time of assessment (32.5% of patients and 52.4% of caregivers were active workers, *χ*^2^ = 18.56, *p* < 0.001). The majority of patients and informal caregivers were Spanish (95.2% and 95.5%, respectively).

### 3.2. Empathic Accuracy in Patients and Informal Caregivers

Differences between self and other reported pain severity and interference (i.e., empathic accuracy) are reported in Table 2. Informal caregivers were good at estimating a patient’s pain severity (*t* = 0.57, *p* = 0.573) and interference (*t* = −0.24, *p* = 0.814). Conversely, patients underestimated the interference of pain in the life of their caregiver (*t* = 2.16, *p* = 0.032). 

Next, we compared patient and caregiver empathic accuracy according to the type of caregiver-patient relationship (Table 3). There were no significant differences in empathic accuracy of patients and caregivers depending on whether the main caregiver was the partner or not (all *p* > 0.050).

### 3.3. Sex Differences in Empathic Accuracy

The results on male-to-female differences in patient and caregiver empathic accuracy are indicated in Table 4. Male and female patients were similarly accurate at estimating the interference of their own pain on the caregiver (*t* = 1.76, *p* = 0.080). This was also the case of male and female caregivers when patient pain interference was the outcome (*t* = 0.85, *p* = 0.394). Conversely, female caregivers were more prone to overestimate their pain severity compared to male caregivers (*t* = 2.18, *p* = 0.030). 

### 3.4. Personality Correlates of Empathic Accuracy and Associations with Health Outcomes

Personality and health associations with empathic accuracy outcomes are presented in Table 5. As noted earlier, four dimensions of empathic accuracy were computed: (i) overall disagreement, which can range from −10 (maximum overestimation) to +10 (maximum underestimation) and is calculated as the difference between self-reported pain severity/interference and the other’s estimate; (ii) disagreement in absolute value, which can take values between 0 and +10 and reflects poor estimation irrespective of the direction of the misestimation; (iii) overestimation, which reflects the amount of individuals for whom disagreement was over 1 SD below the mean of disagreement; and (iv) underestimation, which reflects the amount of individuals for whom disagreement was over 1 SD above the mean of disagreement.

Overall, the patient’s personality was not associated with any form of empathic accuracy (all *p* > 0.01). We found a number of associations between patient health status and patient and caregiver empathic accuracy. Overall, estimation was more difficult as patient status, mostly physical functioning, improved. Specifically, the quality of estimation (i.e., disagreement in absolute value) of patient pain severity (*r* = 0.20; *p* < 0.001), patient pain interference (*r* = 0.29; *p* < 0.001), and caregiver pain interference (*r* = 0.19; *p* = 0.001) worsened as patient physical functioning increased. Similarly, the estimation of patient pain interference (i.e., disagreement in absolute value) was also more challenging as patient overall health status (*r* = 0.16; *p* = 0.008) and mental health (*r* = 0.15; *p* = 0.009) improved. The most frequent judgment error when patient pain status was favorable was overestimation. For instance, good physical functioning was associated with higher overestimation of patient pain severity (*r* = 0.15 *p* = 0.009) and patient pain interference (*r* = 0.17; *p* = 0.005), while general health (*r* = 0.16; *p* = 0.008) and mental health (*r* = 0.19; *p* = 0.002) were positively linked to overestimation of patient pain interference. 

Contrary to patient personality, spouse personality characteristics, notably openness to experience, were related to empathic accuracy. Caregiver openness to experience was associated with a better estimation of the patient’s pain severity (i.e., less disagreement in absolute value; *r* = −0.17; *p* = 0.004), as well as negatively linked to overestimation (*r* = −0.27; *p* < 0.001). The positive association between disagreement in pain ratings and caregiver openness to experience (*r* = 0.20; *p* < 0.001) should be interpreted as showing that caregivers reporting high openness were more likely to underestimate a patient’s pain reports compared to caregivers with a low openness profile, who were more prone to overestimate a patient’s pain severity. In addition to caregiver’s openness, the analyses also revealed that estimation of the interference of pain on the caregivers varied as a function of caregiver agreeableness. Similar to openness, the positive association between disagreement in caregiver pain interference and caregiver agreeableness (*r* = 0.15; *p* = 0.009) should be interpreted as indicating that underestimation of the caregiver’s pain interference was more frequent when caregivers reported high agreeableness, compared to caregivers with a low agreeableness profile, with whom overestimation of pain interference was more frequent. Again, contrary to the patient findings, caregiver health status was not linked to empathic accuracy (all *p* > 0.01).

### 3.5. Multivariate Associations between Patient and Caregiver Predictors and Outcomes

As revealed in the bivariate analyses, estimation of patient and caregiver status was mostly associated with patient health status (i.e., physical functioning, general health, and mental health) and caregiver personality (i.e., openness and, to a lesser extent, agreeableness). Sex differences in empathic accuracy were also revealed in Table 3. Therefore, patient and caregiver sex, patient health status, and caregiver personality (i.e., openness and agreeableness) were included in the regression predicting (i) empathic accuracy towards the patient’s pain severity, (ii) empathic accuracy towards the patient’s pain interference, and (iii) empathic accuracy towards the caregiver’s pain interference. The findings are reported in Table 6. 

Even though different forms of inaccuracy have been calculated in the study (i.e., overall disagreement, disagreement in absolute value, overestimation, and underestimation), we computed the linear multivariate regressions with disagreement in absolute value only (i.e., any form of poor estimation). This decision was based on the fact that this was the type of estimation with which most simultaneous bivariate associations occurred and because this represents well a general measure of poor empathic accuracy irrespective of the direction of such disagreement.

Overall, the explored models contributed significant variance to the prediction of |disagreement| in patient pain status, namely pain severity (*R*^2^ = 5.8%, *F* = 3.57, *p* = 0.001) and pain interference (*R*^2^ = 6.6%, *F* = 3.95, *p* < 0.001), but not in the prediction of caregiver status (*R*^2^ = 1.7%, *F* = 0.10, *p* = 0.102).

Regarding the contribution of predictors on outcomes, the analyses revealed that patient (increased) physical functioning was associated with all study outcomes, namely poor empathic accuracy towards patient’s pain severity (*B* = 0.01, *t* = 2.96, *p* = 0.003, 95% CI = 0.003, 0.015) and interference (*B* = 0.01, *t* = 3.94, *p* < 0.001, 95% CI = 0.006, 0.018) and empathic inaccuracy towards the caregiver’s pain interference status (*B* = 0.01, *t* = 2.61, *p* = 0.009, 95% CI = 0.003, 0.023). Additionally, (increased) caregiver openness was uniquely associated with more accurate estimation of the patient’s pain severity status (*B* = −0.03, *t* = −3.41, *p* < 0.001, 95% CI = −0.045, −0.012).

## 4. Discussion

In this study, we wanted to contribute to the body of literature that examines how social factors are involved in the chronic pain experience. In the current investigation, we explored how patient and caregiver personality correlates to empathic accuracy. Further, we investigated differences in empathic accuracy between partners and other main caregivers and across sex groups, including not only patients but also informal caregivers as recipients of empathic accuracy. First, the study revealed that chronic pain patients are likely to be less accurate in estimating the interference of pain on the others than informal caregivers. Additionally, it showed that empathic accuracy is comparable irrespective of the type of partnership (i.e., couple or not a couple) and indicated that sex differences should be considered in empathic accuracy of pain severity. Finally, the results indicated that patient but not spouse health status is associated with empathic accuracy and, on the contrary, it revealed caregiver but not patient personality correlations with the accuracy of estimations.

Past research has associated poor empathic accuracy and non-empathic spouse responses with deleterious outcomes in the pain patient, such as lower perceived social support, emotional distress, and marital satisfaction [30,35,62], which suggests that empathic accuracy is an important variable to be considered in pain research. Consistent with the present study, there is research to suggest that informal caregivers (i.e., spouses in existing previous studies) are good at estimating the patient’s pain severity [30,31,32] or at least better than other observers, including healthcare professionals [29]. However, contrary to the current investigation findings, there are also a number of other studies that have indicated an overall overestimation of patients’ pain severity and disability [36] and a poor estimation of the patient’s interference of pain [30]. While many factors, including smaller sample sizes compared to the present investigation (84 ≤ n ≤ 109) or differences in sample characteristic (i.e., ethnic or age differences, among others), might influence the discrepancies in empathic accuracy across investigations, an interesting finding in the present study was that caregiver personality, particularly openness to experience, was linked to precision in pain severity estimations. Therefore, it is possible that differences in the distribution of caregiver openness profiles in the studied samples might as well contribute to discrepancies in empathic accuracy of pain severity ratings.

To the best of our knowledge, this is the first investigation to explore personality correlates of empathic accuracy in pain literature. Past research associated empathy with agreeableness, conscientiousness, and openness, but not with neuroticism [63]. The same pattern of associations was found between the FFM of personality and perspective taking, which, similar to empathic accuracy, requires taking another perspective and viewing things from their point of view [64]. While the traditional view of empathy might slightly differ from empathic accuracy in pain research, they both certainly bear important similarities, such as the emphasis on interpersonal perspective-taking [26]. Contrary to the aforementioned literature findings, which revealed that conscientiousness was significantly associated with empathy, our study showed how only openness to experience and agreeableness were related to empathic accuracy. Importantly, the association between openness and empathic accuracy towards the patient’s pain severity status remained significant even after controlling for the contribution of sex and patient health status. These findings are consistent with a recent meta-analysis showing a significant association between openness and interpersonal sensitivity, which suggests that openness to experience might be an important factor associated with empathic accuracy [65]. Openness to experience defines individuals with a great variety of interests, who tend to be curious, original, and sensitive to art. On the contrary, people who scored low in openness are more traditional and conservative in their values. Regarding agreeableness, high scores in this personality trait are obtained by altruist, careful, and cooperative individuals. On the opposite pole, we found hostility, egoism, and dominance [66]. Different to openness, the aforementioned meta-analysis [65] failed to obtain a robust association between agreeableness and interpersonal sensitivity, which would explain why agreeableness was only weakly and not uniquely associated with empathic accuracy in our study. While agreeableness has been argued to be a prosocial personality characteristic [67], there seems to be something unique in open individuals that is important for empathic accuracy and not necessarily shared by agreeable persons. Ultimately, what the present study findings suggest is that the personality profile of informal caregivers should not be ignored due to its potential association with empathic ability. Specifically, in the light of the study findings caregivers characterized by a low openness and, to a lesser extent, low agreeableness profile should be considered as target populations for empathic training in pain settings.

Different to the aforementioned conclusion about caregiver personality, the present study did not reveal a set of personality characteristics in the patient that were associated with empathic accuracy, which suggests that good estimates can be reached irrespective of patient personality characteristics and places the focus on caregiver personality traits in empathic accuracy research. Interestingly, though, associations between empathic accuracy and patient characteristics did emerge for patient health status (i.e., poorer estimation of patient pain severity and patient and caregiver pain interference emerged as patient physical functioning improved and the associations were robust to the inclusion of other covariates). So far, the literature has revealed that spousal accuracy in estimating a patient’s pain severity decreased as patient status (i.e., pain severity, pain interference, and disability) improved [31]. Thus, it is possible that estimating the patient status becomes more challenging when a patient is less physically impaired by pain. Another possibility, though, is that empathic accuracy is perceived as less important when patient status is favorable. Accurately inferring the feelings or states of the pain patient implies an important physical and emotional exhaustion in the caregiver [68], which could be minimized by being imprecise if the situational demands are low (i.e., patient status is better). Due to the non-experimental and cross-sectional nature of the data, it is not possible to conclude whether these associations occurred because better health impacts perceptions of pain, the other way around, or both. Therefore, the proposed interpretations of these findings remain speculative at this stage and require more investigation as the findings may have important implications for care (i.e., whether empathic accuracy should be encouraged or not).

Contrary to patient health status, caregiver health characteristics were not associated with precision of pain severity and interference estimates. There has been scarce research into the caregiver health contributors of empathic accuracy. So far, distressed spouses have been found to overestimate the patient’s pain severity and interference [30,33], but not its physical and psychosocial disability [30]. Our results are in line with the latter study indicating a lack of association between caregiver health status and their empathic accuracy. Further research with larger sample sizes (i.e., sample sizes before the present study completion have been relatively small) and exploring empathic accuracy of important caregivers other than spouses will be needed to clarify whether imprecision is indeed associated with caregiver health status so that this is used to select couples at risk for poor empathic outcomes.

In addition to the study of personality and health correlates of empathic accuracy, important contributions of the present investigation included the study of sex differences in precision and the inclusion of partners and other main caregivers altogether. In relation to sex differences, it has been argued that women are more precise in estimating the physical and emotional states of their counterparts, arguably due to differences in motivation and socialization but not in ability [26,69]. Consistent with this idea, male spouses of chronic pain patients have been found to underestimate the patient’s physical disability, while female spouses have been shown to accurately estimate such disability [36]. Contrary to this idea, another investigation indicated that male and female spouses were comparably good at inferring the care recipient’s pain severity [31] and a recent experimental study revealed that male-to-female differences in accuracy are more likely to be due to stereotypes than to real differences [70]. In our investigation, men and women were very similar at estimating the interference of pain on the other (both when the recipient of empathic accuracy was the patient and the caregiver), which would be in line with research indicating no sex differences in empathic accuracy [31]. If, indeed, motivational reasons are key predictors explaining sex differences in empathy, one possible explanation for male-to-female similarities in judgment is that, in situations where empathy is perceived as important (i.e., in the presence of illness), men and women might become similarly motivated to estimate the status of their counterparts, thus leading to comparable empathic accuracy estimates. Surprisingly, though, our study revealed that female caregivers were more likely to overestimate the severity of a patient’s pain compared to male caregivers. Research has shown that spouse catastrophizing, which is a tendency to worry and focus on the worst possible pain-related scenarios [71], is associated with increased perceptions of patient pain severity [72]. This tendency to ruminate and magnify the threat value of pain is more marked in women [73], which might provide with a tentative explanation for the higher tendency to present increased pain severity estimates of female caregivers in the sample. Again, because of the novelty of these findings and the differences in sample composition, sample size, and together with other methodological differences between the current and past similar research (i.e., measures and constructs evaluated), replication studies will be crucial to obtaining a reliable body of research into the social components into pain research.

As noted in the previous paragraph, the inclusion of caregivers other than spouses is novel in empathic accuracy literature. Past research has already highlighted the important supportive role of informal caregivers other than spouses in pain settings [74]. However, the study of informal caregiver associations with patient outcomes is dominated by research on spouses [27]. Similar to the literature into other chronic conditions [75], the present study indicated that, while spouses of chronic pain patients are likely to play a key role in daily care of pain patients, other actors, particularly children, are also likely to gain prominence in pain research studies. Interestingly, our results evidenced similar empathic accuracy estimates across partners and non-partners, even though both groups are likely to differ in a number of important characteristics (i.e., age, sex distribution, working status, and number of hours spent together, among others). As noted before, it is possible that high motivation to help a person in need might explain that all agents were similarly effective in estimating the patient’s pain status. However, the information available in the present study does not allow us to draw further conclusions and they only provide initial interesting findings in relation to empathic accuracy similarities between partners and other relatives/friends acting as main caregivers of pain patients.

While acknowledging the aforementioned strengths of the present investigation, there are also shortcomings in the study. First, because this is a cross-sectional and observational study, causal and temporary inferences cannot be derived from the results and should be interpreted with caution. The same conclusion applies to the novelty of some of the questions investigated in the study, which will need further replication. Additionally, the patient sample was characterized by experiencing chronic musculoskeletal pain. While this is a frequent practice in pain research into empathic accuracy due to the representativeness of such samples of the populations attending specialized pain clinics [31,33,39], this clearly prevents us from generalizing the findings to specific pain populations. Another limitation of this study is that the list of variables that might be associated with empathic accuracy is far from complete. For instance, time spent together could be an interesting variable to be explored in empathic accuracy research, but our question (“What is the average number of hours you spend with the patient daily?”) was left blank by many caregivers and was misinterpreted by others (i.e., some people reported “24 h” to reflect that they spent all day together, while others did not count sleeping hours or were more conservative and reported less hours despite spending all day together). Another variable that has been linked with empathy in the literature but has not been investigated in the present study is depression [36]. An additional shortcoming has been that we could not visually confirm that patients and caregivers indeed completed to the questionnaires separately. While we did visually confirm that the questionnaires were returned in separate, sealed envelopes and we observed differences in patient-to-caregiver estimates (i.e., patients slightly underestimated the caregivers’ status), which would suggest that the measures were indeed completed separately, we cannot firmly conclude this. As a final remark, the reader should note that all effect sizes were small (i.e., significant correlations and Cohen’s *d* were all <0.03) and the explored multivariate models contributed little variance to the prediction of empathic accuracy (i.e., less than 7%), so the strength of the presented associations and mean differences should not be overestimated.

While acknowledging the aforementioned shortcomings, it should be noted that the present investigation explored important contextual, bottom-up and top-down correlates of empathic accuracy described in the empathic accuracy model of pain by Dr. Goubert and her team [28]. Specifically, we found support for top-down mechanisms (i.e., the observer’s openness to experience and, to a lesser extent, agreeableness) and bottom-up factors (i.e., patient health status, mostly physical functioning) associations with empathic accuracy. Conversely, empathic accuracy differences were not revealed as a function of the type of relationship (i.e., contextual factor). As noted earlier, replicating these findings and exploring different sets of mechanisms will be fundamental to provide more robust evidence for the model proposed by Goubert et al. [28]. However, the present study findings make a significant contribution into the literature on empathic accuracy and provide further support for the need of pain models that account for both personal and interpersonal mechanisms that are activated to respond to pain.

## 5. Conclusions

This study contributes to the existing literature on the social context of chronic pain in a number of ways. First, because it revealed that empathic accuracy, which has been argued to be a protective factor for patient well-being, should not only consider patients as recipients of empathy but also caregivers. Next, because it highlighted that sex differences in empathic accuracy might take unexpected directions when considering larger sample sizes and indicated that informal caregivers other than spouses should be incorporated in research into social contributors of the pain experience. Last but not least, an important finding in the present investigation was that caregiver characteristics (i.e., low openness and low agreeableness) and patient health status (notably increased physical functioning) might be significant factors associated with difficulties in empathic accuracy in pain settings. Ultimately, the importance of these findings lies in their applicability in clinical settings. For instance, in light of our results, target populations for empathic training would be non-agreeable and non-open caregivers, together with dyads in which the patient status is favorable, irrespective of the type of relationship between patients and caregivers (i.e., couples or non-couples). Thus, the present study findings might help guide biopsychosocial interventions in a more effective manner by providing a set of recommendations to be included in the social part of multidimensional treatments for pain management.

## Figures and Tables

**Table 1 medicina-55-00539-t001:** Sociodemographic characteristics of the sample (*n* = 292).

	Patient *n* (%)	Caregiver *n* (%)
Sex		
Women	195 (66.8)	149 (51.0)
Men	97 (33.2)	143 (49.0)
Age Mean (*SD*)	59.35 (15.76)	53.52 (14.69)
Marital Status		
In a relationship	202 (69.2)	226 (77.4)
Not in a relationship	90 (30.8)	66 (22.6)
Educational level		
<12 years	218 (74.7)	191 (65.4)
>12 years	74 (25.3)	101 (34.6)
Employment		
Working	95 (32.5)	153 (52.4)
Not working	197 (67.5)	139 (47.6)
Nationality		
Spanish	278 (95.2)	279 (95.5)
Non-Spanish	14 (4.8)	13 (4.5)
Relationship		
Partner/Spouse	-	200 (68.5)
Other ^1^	-	92 (31.5)

^1^ Other = Child (20.2%), sibling (4.1%), parent (2.7%), friend (1.0%), grandchild (0.7%), nephew (0.7%), mother-in-law (0.3%), aunt (0.3%), and child-in-law (0.3%).

**Table 2 medicina-55-00539-t002:** Pain severity and interference characteristics and differences in empathic accuracy between patients and caregivers (*n* = 292).

	Patient Report Mean (*SD*)	Caregiver Report Mean (*SD*)	*t*	*p*	*d*
Patient pain severity	7.80 (1.66)	7.87 (1.57)	0.57	0.573	0.04
Patient pain interference	8.15 (1.58)	8.12 (1.59)	−0.24	0.814	0.02
Caregiver pain interference	7.02 (2.39)	7.42 (2.09)	2.16	0.032	0.18

**Table 3 medicina-55-00539-t003:** Empathic accuracy as a function of partner relationship (*n* = 292).

	Partner/Spouse *n* = 200 Mean (*SD*)	Not Partner *n* = 92 Mean (*SD*)	*t*	*p*	*d*
Disagreement in patient pain severity	−0.16 (1.40)	0.10 (1.37)	−1.44	0.150	0.19
Disagreement in patient pain interference	<0.01 (1.52)	0.10 (1.30)	−0.54	0.593	0.06
Disagreement in caregiver pain interference	0.39 (2.19)	0.43 (2.48)	−0.17	0.863	0.02

Note. Disagreement is calculated by subtracting the other’s estimation to self-report (i.e., patient’s report of patient pain severity-caregiver’s report of patient pain severity). Values close to “0” indicate a good estimation.

**Table 4 medicina-55-00539-t004:** Empathic accuracy as a function of sex (*N* = 292).

	Male Estimator Mean (*SD*)	Female Estimator Mean (*SD*)	*t*	*p*	*d*
Estimator = caregiver	*n* = 143	*n* = 149			
Disagreement in patient pain severity	−0.10 (1.43)	−0.25 (1.34)	2.18	0.030	0.11
Disagreement in patient pain interference	0.10 (1.62)	−0.04 (1.27)	0.85	0.394	0.10
Estimator = patient	*n* = 97	*n* = 195			
Disagreement in caregiver pain interference	0.73 (2.01)	0.24 (2.39)	1.76	0.080	0.22

Note: Disagreement is calculated by subtracting the other’s estimation to self-report (i.e., patient’s report of patient pain severity-caregiver’s report of patient pain severity). Values close to “0” indicate a good estimation.

**Table 5 medicina-55-00539-t005:** Means and standard deviations of study variables and Pearson bivariate associations between empathic accuracy measures and patient and caregiver personality characteristics and health status.

		Patient Personality					Patient Health Status			Caregiver Personality					Caregiver Health Status		
	Mean (*SD*)/*n* (%)	N	E	O	A	C	PF	GH	MH	N	E	O	A	C	PH	GH	MH
Patient pain severity																	
Disagreement	−0.08 (1.39)	0.06	0.06	−0.01	0.02	0.01	−0.07	−0.11	−0.15	−0.06	0.06	0.20 **	−0.07	0.02	0.08	0.06	0.05
|Disagreement|	0.92 (1.04)	0.06	−0.01	0.04	0.01	−0.06	0.20 **	0.12	0.07	0.12	−0.10	−0.17 *	−0.04	−0.04	−0.03	−0.01	−0.01
Overestimation	31 (10.6%)	−0.03	0.04	−0.07	0.04	−0.05	0.15 *	0.13	0.13	0.10	−0.01	−0.27 **	0.04	0.01	−0.07	−00.04	−0.04
Underestimation	26 (8.9%)	0.12	−0.01	−0.06	0.01	−0.06	0.06	−0.07	−0.06	0.02	−0.01	−0.01	−0.05	0.02	−0.05	.05	0.07
*Patient pain interference*																	
Disagreement	0.03 (1.45)	0.07	0.01	0.08	−0.01	0.04	0.02	−0.06	−0.11	−0.03	0.07	0.10	−0.04	−0.02	0.01	−0.03	−0.03
|Disagreement|	0.98 (1.07)	−0.03	0.06	0.06	0.03	0.08	0.29 **	0.16 *	0.15 *	0.10	−0.03	0.01	−0.05	−0.04	0.05	0.05	0.04
Overestimation	32 (11%)	−0.12	0.06	−0.06	0.06	0.05	0.17 *	0.16 *	0.19 *	0.10	−0.08	−0.09	−0.04	−0.04	0.06	0.05	0.05
Underestimation	36 (12.3%)	0.06	0.01	0.07	−0.02	0.04	0.16 *	0.07	−0.03	0.11	0.01	0.03	−0.07	−0.08	0.01	−0.02	−0.08
*Caregiver pain interference*																	
Disagreement	0.40 (2.28)	−0.05	<0.01	−0.04	0.05	<0.01	0.10	−0.02	0.14	0.11	−0.04	−0.08	0.15 *	0.05	−0.11	−0.08	−0.07
|Disagreement|	1.54 (1.73)	0.01	−0.02	0.09	−0.03	0.04	0.19 *	0.10	0.09	0.01	−0.01	0.02	0.05	0.04	0.02	0.02	0.06
Overestimation	40 (13.7%)	0.05	−0.01	0.03	−0.05	0.04	0.03	0.07	−0.09	−0.14	0.05	0.10	−0.11	−0.04	0.13	0.09	0.11
Underestimation	46 (15.8%)	−0.09	0.01	−0.02	0.05	0.06	0.15	0.10	0.14	0.09	−0.01	−0.04	0.06	0.03	−0.04	−0.05	−0.01
Mean (SD)		24.66 (8.88)	26.28 (7.73)	22.50 (6.90)	31.82 (6.27)	30.97 (7.34)	32.77 (23.73)	34.23 (18.68)	50.23 (19.76)	19.92 (8.45)	27.82 (7.67)	24.81 (7.16)	30.98 (5.92)	32.02 (6.94)	77.11 (28.44)	61.40 (23.44)	66.60 (20.57)

Note. For “over/underestimation” variables values in the first column represent “frequencies (percentage)” of individuals who overestimate or underestimate, respectively (0 = ”Estimates well, ±1SD from the disagreement mean” and 1 =”Over/Underestimates, over ±1SD from the disagreement mean”). The symbol “||” reflects absolute values. N = Neuroticism; E = Extraversion; O = Openness to experience; A = Agreeableness; C = Conscientiousness; PF = Physical Functioning; GH = General Health; MH = Mental Health. ** *p* < 0.001, * *p* < 0.01.

**Table 6 medicina-55-00539-t006:** Linear multivariate regression analyses predicting empathic accuracy from patient and caregiver characteristics.

	|Disagreement| in Patient Pain Severity			|Disagreement| in Patient Pain Interference			|Disagreement| in Caregiver Pain Interference		
	*B* (95% CI)	SE	*t*	*B* (95% CI)	SE	*t*	*B* (95% CI)	SE	*t*
Patient sex	0.08 (−0,21, 0.37)	0.15	0.56	−0.03 (−0.33, 0.27)	0.15	−0.20	−0.10 (−0.59, 0.40)	0.25	−0.38
Caregiver sex	0.08 (−0.22, 0.38)	0.15	0.52	0.05 (−0.26, 0.36)	0.16	0.32	0.07 (−0.44, 0.59)	0.26	0.27
Patient physical functioning	0.01 (<0.01, 0.02)	<.01	2.96 *	0.01 (0.01, 0.02)	<0.01	3.94 **	0.01 (<0.01, 0.02)	0.01	2.61 *
Patient general health	<0.01 (−0.01, 0.01)	<0.01	0.87	<0.01 (−0.01, 0.01)	<0.01	0.14	<0.01 (−0.01, 0.01)	0.01	0.09
Patient mental health	>−0.01 (−0.01, 0.01)	<0.01	−0.34	<0.01 (−0.01, 0.01)	<0.01	0.72	<0.01 (−0.01, 0.01)	0.01	0.25
Caregiver openness	−0.03 (−0.05, −0.01)	0.01	−3.41 **	>−0.01 (−0.02, 0.01)	0.01	−0.39	<0.01 (−0.03, 0.03)	0.01	0.09
Caregiver agreeableness	−0.01 (−0.03, 0.01)	0.01	−0.74	−0.01 (−0.03, 0.01)	0.01	−0.66	0.02 (−0.02, 0.05)	0.02	1.00
*F*	3.57 *			3.95 **			0.10		
*R* ^2^	5.8%			6.6%			1.7%		

** *p* < 0.001, * *p* < 0.01. *R*^2^ is adjusted. Betas are unstandardized. Higher |disagreement| should be interpreted as poorer empathic accuracy.

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
