# Peer review of "Empathic Accuracy in Chronic Pain: Exploring Patient and Informal Caregiver Differences and Their Personality Correlates"

_medicina, 2019, doi:10.3390/medicina55090539_

Round 1

Reviewer 1 Report

Abstract:

In the abstract, the following sentence is not clear: « patients were less accurate at estimating the interference of pain on their counterparts than informal caregivers ». I do not understand: than informal caregivers at what?

The next sentence is not clear either: “empathic accuracy did not differ across types of relationship…”. Empathic accuracy on what?

Introduction: ok

Measures

Pain severity and interference: What was the time range of the question? For example, was it “during the last week? In the last month?”  Other time range?

Results

Results are clearly presented. My main regret is the lack of a multivariate model. You have enough observations to perform it. Once you have selected the main predictors with the mean of your first analyses reported in the current version, you should include in a multivariate model the following predictors – patient vs caregiver report, patient health status PF, GH, MH, caregiver personality O and estimator gender – to explain your three outcomes (3 multivariate analyses).

Discussion: interesting discussion

Author Response

Dear Reviewer #1,

Thank you very much for taking the time to read our manuscript and for the input given. Please find below our responses to the comments made on our manuscript.

Abstract:

In the abstract, the following sentence is not clear: « patients were less accurate at estimating the interference of pain on their counterparts than informal caregivers ». I do not understand: than informal caregivers at what?

Response: Indeed, the sentence was unclear. It now reads: “Patients were relatively inaccurate at estimating the interference of pain on their counterparts (t=2.16; p=.032), while informal caregivers estimated well the patient’s status (all differences p>.05).”

The next sentence is not clear either: “empathic accuracy did not differ across types of relationship…”. Empathic accuracy on what?

Response: Again, we agree that the sentence needed rewriting. The current version states the following: “Empathic accuracy on the patients’ and caregivers’ status did not differ across types of relationship (i.e., couple or other; all differences p>.05).”

Introduction: ok

Response: Thanks a lot.

Measures

Pain severity and interference: What was the time range of the question? For example, was it “during the last week? In the last month?”  Other time range?

Response: Thanks for noticing this. We now specify that pain severity and interference were evaluated “in the last 24h”. This time range is the one used in the Brief Pain Inventory, a widely used measure of pain status.

Results

Results are clearly presented. My main regret is the lack of a multivariate model. You have enough observations to perform it. Once you have selected the main predictors with the mean of your first analyses reported in the current version, you should include in a multivariate model the following predictors – patient vs caregiver report, patient health status PF, GH, MH, caregiver personality O and estimator gender – to explain your three outcomes (3 multivariate analyses).

Response: We completely agree. We have now computed the proposed multivariate regressions (see Table 6 and the results section for the changes made, which have also been highlighted throughout the discussion).

Discussion: interesting discussion

Response: Thanks a lot.

We want to thank you again for the comments made, which have helped improve the quality of the text. We hope to have addressed the issues raised.

We look forward to your comments.

Kind regards,

Carlos

Reviewer 2 Report

I’ve now had a chance to review the article entitled, “Empathic accuracy in chronic pain: Exploring patient and informal caregiver differences and their personality correlates.” This research examines the concordance between chronic pain patients’ pain (severity, interference and perceived impact) and informal caregivers’ perception of pain (severity, interference, and actual impact). This research fills important gaps in the literature by examining psychosocial correlates of what the others call empathic accuracy with an impressive sample of chronic pain patients and their caregivers. However, the paper lacks background literature and clarity in terminology that with a major revision, can be addressed.

Introduction:

First, while the paper is grounded in William Ickes’ social psychology empathic accuracy literature, what the paper lacks is a grounding in the psychology research around pain perception and assessment. For example, Goubert’s empathic pain accuracy model (Goubert, L., Craig, K. D., Vervoort, T., Morley, S., Sullivan, M. J. L., de CAC, W., ... & Crombez, G. (2005). Facing others in pain: the effects of empathy. Pain118(3), 285-288) would be important to include in the introduction as is a published meta-analysis on this topic of patient-other pain assessment accuracy (Ruben, Mollie A., Danielle Blanch-Hartigan, and Jillian C. Shipherd. To know another’s pain: A meta-analysis of caregivers’ and healthcare providers’ pain assessment accuracy. Annals of Behavioral Medicine 52.8 (2018): 662-685). This meta-analysis should also be reintegrated into the discussion comparing findings.

While the authors lay out the complexity of pain (multidimensional and biopsychosocial), what the introduction is lacking is a discussion about the criterion since the authors use self-report as the criterion of accuracy but there could be and perhaps should be many other factors that go into understanding the pain experience from a caregivers’ perspective.

The introduction also lacks a discussion about accuracy as this has been operationalized in many ways and includes under/over estimation and correlational accuracy. These different methods of assessing accuracy matter as one could distinguish different levels of pain at different time points but have a tendency to always underestimate their loved one’s pain.

Finally, the introduction is lacking general statements about why accuracy matters. Evolutionarily speaking- why would perceiving others’ pain accurately matter for that person and the sufferer? When this is chronic pain rather than acute pain- doesn’t accuracy have the same consequence? Is it possible that accurately perceiving pain could reinforce pain behaviors? These are all points that should be addressed in the intro and/or discussion. Finally, why might perceiving the impact a person’s pain has on their loved one matter? Wouldn’t it be adaptive to not perceive that effect if they needed to put their resources into their own pain and suffering?

On p. 3, you mention personality characteristics but don’t give an overview of what these are- there are hundreds of personality characteristics that could be measured. An overview in the intro of the significance of the big five is important here.

Throughout the paper, empathic accuracy is used as the terminology but there are really three different types of empathic accuracies assessed specific to pain. It would be helpful to change the terminology to empathic pain accuracy and specify whether you’re talking about pain intensity/severity, interference, or caregiver impact. Hypotheses should also be specific to these three types of empathic pain accuracy rather than the one concept.

Methods:

How did you ensure that patients and caregivers completed the questionnaires separately?

How did you verify that caregivers were the “main informal caregiver”?

P5, line 201- instead of the arbitrary p < .01, you should use Bonferroni corrections to assess p-values. If this results in rare occurrences of significant values, that shouldn’t matter since you report effect sizes throughout.

Results

More should be made about the overestimation of pain severity and interference when better physical functioning as this has very important implications for the patient. Is it that because they’re treated in this way that they are actually reporting better health or is the better health that is impacting perceptions of pain? You could imagine this relationship going either way and has important implications for the care of those with chronic pain as one could reinforce pain behaviors and poorer health status when perceiving accurate or more pain than actually exists.

Did you measure patient level of depression? This has been linked to accuracy in past work (as you suggest) and is comorbid with chronic pain- could this explain some of the inability to infer impact on caregiver?

Discussion

While not specific to pain, a meta-analysis on correlates of interpersonal sensitivity (Hall, J. A., Andrzejewski, S. A., & Yopchick, J. E. (2009) Psychosocial correlates of interpersonal sensitivity: A meta-analysis. Journal of Nonverbal Behavior33(3), 149-180) shows a similar pattern for openness but not agreeableness.

Line 365- you mention that women are better at perceiving emotional and physical states due to differences in motivation but there is also literature to suggest this is part of socialization and that men are actually better at perceiving pain than women.

Minor edits

Page 3, Line 127, “recurrent pain forover”

Page 7 Line 267: overtimation- to overestimation

Line 422- existent to existing?

Author Response

Dear Reviewer #2,

Thank you very much for taking the time to read our manuscript and for the input given. Please find below our responses to the comments made on our manuscript.

I’ve now had a chance to review the article entitled, “Empathic accuracy in chronic pain: Exploring patient and informal caregiver differences and their personality correlates.” This research examines the concordance between chronic pain patients’ pain (severity, interference and perceived impact) and informal caregivers’ perception of pain (severity, interference, and actual impact). This research fills important gaps in the literature by examining psychosocial correlates of what the others call empathic accuracy with an impressive sample of chronic pain patients and their caregivers. However, the paper lacks background literature and clarity in terminology that with a major revision, can be addressed.

Introduction:

First, while the paper is grounded in William Ickes’ social psychology empathic accuracy literature, what the paper lacks is a grounding in the psychology research around pain perception and assessment. For example, Goubert’s empathic pain accuracy model (Goubert, L., Craig, K. D., Vervoort, T., Morley, S., Sullivan, M. J. L., de CAC, W., ... & Crombez, G. (2005). Facing others in pain: the effects of empathy. Pain118(3), 285-288) would be important to include in the introduction as is a published meta-analysis on this topic of patient-other pain assessment accuracy (Ruben, Mollie A., Danielle Blanch-Hartigan, and Jillian C. Shipherd. To know another’s pain: A meta-analysis of caregivers’ and healthcare providers’ pain assessment accuracy. Annals of Behavioral Medicine 52.8 (2018): 662-685). This meta-analysis should also be reintegrated into the discussion comparing findings.

Response: Thanks for this important comment. We completely agree that the introduction has clearly benefited from the inclusion of both references and a more detailed description of the literature based on the model proposed by Dr. Goubert’s team. We have made changes in the introduction (e.g., “A recent meta-analysis also indicated that informal caregivers are more accurate than physicians and revealed that pain is more frequently underestimated in male patients [29]” and “According to Goubert et al. [28], empathic accuracy is determined by contextual factors (e.g., type of relationship and affinity), top-down mechanisms (e.g., the observer’s personality characteristics and past learning experiences), and bottom-up influences (e.g., the observed person’s characteristics). In the present study, contextual, top-down, and bottom-up mechanisms will be included”), as well as in the discussion to reflect this (“While acknowledging the aforementioned shortcomings, it should be noted that the present investigation explored important contextual, bottom-up, and top-down correlates of empathic accuracy described in the empathic accuracy model of pain by Dr. Goubert and her team [28].  Specifically, we found support for top-down mechanisms (i.e., the observer’s openness to experience and, to a lesser extent, agreeableness) and bottom-up factors (i.e., patient health status, mostly physical functioning) associations with empathic accuracy. Conversely, empathic accuracy differences were not revealed as a function of the type of relationship (i.e., contextual factor). As noted earlier, replicating these findings and exploring different sets of mechanisms will be fundamental to provide more robust evidence for the model proposed by Goubert et al. [28]. However, the present study findings make a significant contribution into the literature on empathic accuracy and provide further support for the need of pain models that account for both personal and interpersonal mechanisms that are activated to respond to pain)”.

While the authors lay out the complexity of pain (multidimensional and biopsychosocial), what the introduction is lacking is a discussion about the criterion since the authors use self-report as the criterion of accuracy but there could be and perhaps should be many other factors that go into understanding the pain experience from a caregivers’ perspective.

Response: We completely agree that empathic accuracy is only one way of exploring the pain experience. We have now better conceptualized the criterion in relation to other criteria (“Research into empathic accuracy can be included in a tradition of studies exploring the experience of pain from a caregiver’s perspective, which includes not only spouse appraisals about the other’s status (i.e., empathic accuracy), but also perceptions about one’s status (i.e., caregiving burden) and emotional and behavioral reactions to pain (e.g., validating or critical responses) [27,28]”).

The introduction also lacks a discussion about accuracy as this has been operationalized in many ways and includes under/over estimation and correlational accuracy. These different methods of assessing accuracy matter as one could distinguish different levels of pain at different time points but have a tendency to always underestimate their loved one’s pain.

Response: Thanks for this comment. We completely agree that this was necessary in the introduction. We have included a paragraph describing this (“Two types of empathic accuracy have been traditionally tested, both relying on self-reports. On the one hand, correlational accuracy indicates how well self-reported patient pain is associated with the observer’s perception of the patient’s pain. This perspective is frequently used when the patients’ and the observers’ ratings are based on different response scales, but its interest is limited because it does not provide information about the size and the direction of inaccuracy. An alternative to this is the paired comparison accuracy, which evaluates the extent to which observers are accurate at estimating the other’s status (i.e., overestimation and underestimation) [29]. As noted earlier, this requires the use of the same response scale in patients and observers, but provides more detailed information about empathic accuracy. Thus, the latter will be preferred in the present investigation”).

Finally, the introduction is lacking general statements about why accuracy matters. Evolutionarily speaking- why would perceiving others’ pain accurately matter for that person and the sufferer? When this is chronic pain rather than acute pain- doesn’t accuracy have the same consequence? Is it possible that accurately perceiving pain could reinforce pain behaviors? These are all points that should be addressed in the intro and/or discussion. Finally, why might perceiving the impact a person’s pain has on their loved one matter? Wouldn’t it be adaptive to not perceive that effect if they needed to put their resources into their own pain and suffering?

Response: Indeed, stronger arguments for the importance of empathic accuracy were required. We have added this in the final paragraph of the introduction (“Including empathic accuracy training in such interventions would be beneficial for a number of reasons. Empathic accuracy is argued to lead to prosocial and support-related behaviors [45], proximity feelings in a relationship [46], and adaptive solution of conflicts in romantic relationships [47]. In fact, empathic accuracy has been argued to have a survival value for the patient as it would enhance assistance from others in the presence of pain (i.e., affective and behavioral responses) [28,31]. On the contrary, spouse poor estimation might lead to inadequate support by the caregiver [48], which might ultimately exert a negative impact on the patient’s ability to deal with pain [32]. By exploring contextual, bottom-up, and bottom-down correlates of empathic accuracy, the present study aims at contributing to this important literature on social factors involved in the chronic pain experience.”).

On p. 3, you mention personality characteristics but don’t give an overview of what these are- there are hundreds of personality characteristics that could be measured. An overview in the intro of the significance of the big five is important here.

Response: We completely agree with this comment. We have included this overview in the introduction (“While many personality characteristics exist, we chose to evaluate the five core dimensions of the Five Factor Model of personality because this model is cross-culturally robust and has been repeatedly and reliably associated with important study variables (i.e., health status and empathy) across different populations [42,43], including chronic pain [44]”).

Throughout the paper, empathic accuracy is used as the terminology but there are really three different types of empathic accuracies assessed specific to pain. It would be helpful to change the terminology to empathic pain accuracy and specify whether you’re talking about pain intensity/severity, interference, or caregiver impact. Hypotheses should also be specific to these three types of empathic pain accuracy rather than the one concept.

Response: Thanks for this comment. We have proceeded as suggested throughout the text.

Methods:

How did you ensure that patients and caregivers completed the questionnaires separately? How did you verify that caregivers were the “main informal caregiver”?

Response: Indeed, a more detailed explanation on both topics was required. We now provide further information about both comments in the methods section (“We verified that caregivers were indeed the main informal caregivers when the patient or both returned the questionnaires during the medical appointment. Because the information letter was very clear at this stage, all patients confirmed that their main caregiver responded to the questionnaires. We also verbally confirmed that the questionnaires were completed separately and ensured that the questionnaires were returned in separate and sealed envelopes.”). Additionally, we acknowledge the fact that we are unable to ensure that questionnaires were completed separately in the limitation section (“An additional shortcoming has been that we could not visually confirm that patients and caregivers indeed completed to the questionnaires separately. While we did visually confirm that the questionnaires were returned in separate, sealed envelopes and we observed differences in patient-to-caregiver estimates (i.e., patients slightly underestimated the caregivers’ status), which would suggest that the measures were indeed completed separately, we cannot firmly conclude this”).

P5, line 201- instead of the arbitrary p < .01, you should use Bonferroni corrections to assess p-values. If this results in rare occurrences of significant values, that shouldn’t matter since you report effect sizes throughout.

Response: Thanks for noticing this. This was an omission from our side. We did calculate the alpha level based on the Holm-Bonferroni sequential correction. We now state it specifically in the methods section (“The new alpha level was calculated using a Holm-Bonferroni sequential correction, a less restrictive correction than the original Bonferroni correction [59])”.

Results

More should be made about the overestimation of pain severity and interference when better physical functioning as this has very important implications for the patient. Is it that because they’re treated in this way that they are actually reporting better health or is the better health that is impacting perceptions of pain? You could imagine this relationship going either way and has important implications for the care of those with chronic pain as one could reinforce pain behaviors and poorer health status when perceiving accurate or more pain than actually exists.

Response: we completely agree that this is an important finding. We have conducted additional (multivariate) analyses to further explore whether this association was retained even after controlling for patient and caregiver sex characteristics, other patient health outcomes, and caregiver personality. Indeed, the analyses revealed that this association between patient physical functioning and poor estimation is quite robust. While our data prevents us from drawing any further causal conclusions about such associations, we have provided some additional discussion about this in the corresponding section.

Did you measure patient level of depression? This has been linked to accuracy in past work (as you suggest) and is comorbid with chronic pain- could this explain some of the inability to infer impact on caregiver?

Response: Indeed, depression has been linked with low empathic accuracy (i.e., spouse depression as predictor of low empathic accuracy towards the patient). In our study, the only measure of mental well-being was the “mental health” scale in the Short Form-36. We did not control for additional measures of depression such as the BDI-II nor conduct any formal diagnosis of depression. We will acknowledge this in the limitations section (“Another variable that has been linked with empathy in the literature but has not been investigated in the present study is depression [36]”).

Discussion

While not specific to pain, a meta-analysis on correlates of interpersonal sensitivity (Hall, J. A., Andrzejewski, S. A., & Yopchick, J. E. (2009) Psychosocial correlates of interpersonal sensitivity: A meta-analysis. Journal of Nonverbal Behavior33(3), 149-180) shows a similar pattern for openness but not agreeableness.

Response: Thank for very much for the recommendation. Indeed, the paper is very adequate for our text. We have included further discussion in the manuscript in relation to the proposed meta-analysis (“Importantly, the association between openness and empathic accuracy towards the patient’s pain severity status remained significant even after controlling for the contribution of sex and patient health status. These findings are consistent with a recent meta-analysis showing a significant association between openness and interpersonal sensitivity, which suggests that openness to experience might be an important factor associated with empathic accuracy [65]” and “Different to openness, the aforementioned meta-analysis [65] failed to obtain a robust association between agreeableness and interpersonal sensitivity, which would explain why agreeableness was only weakly and not uniquely associated with empathic accuracy in our study. While agreeableness has been argued to be a prosocial personality characteristic [67], there seems to be something unique in open individuals that is important for empathic accuracy and not necessarily shared by agreeable persons”).

Line 365- you mention that women are better at perceiving emotional and physical states due to differences in motivation but there is also literature to suggest this is part of socialization and that men are actually better at perceiving pain than women.

Response: This is an excellent point. In the mentioned paragraph, we were mentioning some literature findings and assumptions (which we do not share, based on recent research on this topic). We have included some additional text to better reflect this idea (“In relation to sex differences, it has been argued that women are more precise in estimating the physical and emotional states of their counterparts, arguably due to differences in motivation and socialization but not in ability [26,69]… Contrary to this idea, … and a recent experimental study revealed that male-to-female differences in accuracy are more likely to be due to stereotypes than to real differences [70]”).

Minor edits

Page 3, Line 127, “recurrent pain forover”

Page 7 Line 267: overtimation- to overestimation

Line 422- existent to existing?

Response: thanks for noticing these mistakes.

We want to thank the reviewer again for the comments made, which have helped improve the quality of the text. We hope to have addressed the issues raised.

We look forward to your comments.

Kind regards,

Carlos

Round 2

Reviewer 2 Report

Thank you to the authors for their attention to revisions. I have no other changes. This is a strong paper and an important contribution to the literature.